



# Linkage of tropical glaciation to supercontinents: a thermodynamic closure model

Hsien-Wang Ou[1]

[1] Lamont-Doherty Earth Observatory, Columbia University, Palisades, NY 10964, USA

*Correspondence to*: H.-W. Ou (hsienou0905@gmail.com)

**Abstract.** Precambrian tropical glaciations pose a significant challenge to our understanding of Earth's climate. A popular explanation invokes runaway ice-albedo feedback leading to "iceball earth", an extreme state conflicting however with the sedimentary evidence of an open ocean and active hydrological cycle. We point out flawed physics of the runaway scenario, which overlooks potency of the ocean heat transport in deterring the perennial sea ice. Nor is frozen ocean needed for tropical

glaciation as the latter requires only that the tropical land be cooled to below the marking temperature of the glacial margin, which is necessarily above the freezing point to counter the yearly accumulation. Since tropical glaciations generally coincide with Precambrian supercontinents, we posit that it is their blockage of the brighter tropical sun that causes the required cooling. To test this hypothesis, we formulate a minimal two-box model, which is nonetheless thermodynamically closed and yields lowering tropical/polar temperatures with increasing tropical land, whose crossings of the glacial marking temperature would

divide non/polar/pan-glacial regimes—the last being characterized by tropical glaciation abutting an open ocean. Given the observed chronology of paleogeography, our theory may provide a unified account of the faint-young-sun paradox, Precambrian tropical glaciations and glacio-epochs through Earth's history.

## 1 Introduction

Cryogenian glacio-epochs stand out in that glaciers have advanced to the sea level in the tropics (Chumakov & Elston, 1989;

Sohl et al., 1999; Evans, 2003; Kilner et al., 2005), a pan-glacial state coined "snowball earth" by Kirschvink (1992) without implicating a frozen ocean. Invoking runaway ice-albedo feedback and peculiar glacial sequence, Hoffman et al. (1998) proposes an extreme interpretation of the tropical glaciation by way of an ice-covered planet, which we shall refer to as "iceball earth" for distinction (Harland, 2007)—avoiding the misnomer of "snow" over a global frozen ocean. The iceball earth hypothesis has largely been refuted by sedimentary evidence: the putative tell-tale signs of cap carbonates and iron formation

turn out to be nonunique to a global frozen ocean (Kennedy et al., 2001; Jiang et al., 2003; Le Guerrouè et al., 2006; Fairchild & Kennedy 2007; Etinne et al., 2007; Eyles, 2008) and dynamic glacial advance/retreat indicates an active hydrological cycle predicated on an open ocean (Christie-Blick et al., 1999; Condon et al., 2002; Leather et al., 2002; Rieu et al., 2007; Chumakov, 2008; Allen & Etienne, 2008). Given extensive literature on the glacial sedimentology, our focus is on the physics of the tropical glaciation, and in this introduction, we shall point out flawed physics of past models, pertaining particularly to glacial



margin, ocean heat transport (OHT), $pCO_2$ and cloud in successive paragraphs below, and preview their possible remedies in our model.

We begin with the energy-balance model, a progenitor of the runaway ice-albedo feedback, which presupposes a common land/sea-ice line taggable by temperature (Budyko, 1969)—overlooking a fundamental difference of the two media in that the ocean can transport heat. While one may tag the land-ice margin by the summer surface-air temperature (SAT), which controls

the ablation, this temperature nonetheless should be above the freezing point to counter the yearly accumulation, and since summer isotherms are predominantly zonal (Sect. 3), the glacial margin thus should abut an open ocean. This argument is consistent with the present Greenland ice sheet protruding into the open North Atlantic and the Laurentide ice sheet of the last ice age extending into the subtropics, and it is the reason that climate models incorporating the ice-sheet dynamics have produced a more expansive land ice (Hyde et al., 2000). By extension, the land ice would advance into the tropics so long as

the tropical land is cooled to below this marking temperature—without a frozen ocean premised in iceball earth (Yang et al., 2012).

For the sea ice, on the other hand, a freezing-point surface provides only a necessary condition for ice formation, which can be sustained (perennial) only if the OHT may not overcome the yearly cooling. Here we come upon an Archilles heel in climate modeling: OHT obviously cannot be properly captured by slab, mixed-layer or diffusive ocean models (Jenkins &

Smith, 1999; Chandler & Sohl, 2000; Bice et al., 2000; Poulsen et al., 2001; Donnadieu et al., 2004; Braun et al., 2022; Hörner et al., 2022), but even in (coarse-grained) primitive-equation models, OHT depends critically on diapycnal diffusivity, which is in effect a free parameter finely tuned to produce the present climate (Bendtsen, 2002; Rahmstorf et al., 2005) and, being a free parameter, there is no reason that the tuned value should apply to a vastly different paleoclimate. We attribute this nonclosure to coarse-graining of the meridional overturn circulation (MOC), which takes the form of a laminar overturning

cell, whereas the actual one is composed partly of random eddies (Lozier, 2010) to be subjected to probability laws of the nonequilibrium thermodynamics (NT). Resolving eddies obviously poses a daunting challenge to the numerical simulation of glacio-epochs, which points to a palpable advantage of our theoretical construct unrestrained by computing resources—not to mention its faculty in elucidating the underlying physics. We should note that even rudimentary treatment of OHT by aforementioned models has significantly curbed the sea-ice advance, and some simulations of the runaway scenario are actually

artifacts of an unbalanced initial state (Voigt et al., 2010; Yang et al., 2012) when OHT simply has no time to act in deterring the sea ice.

Besides the need to include ice-sheet dynamics and OHT, a common yet questionable practice of climate modelling is in prescribing internal properties of the climate system to probe its response. A prominent example is $pCO_2$ (Pollard & Kasting, 2004; Pierrehumbert et al., 2011), which would feedback negatively on temperature to equilibrate over a million-year (Walker

et al., 1981; Brady & Caroll, 1994; Berner & Caldeira, 1997; Godderis & Donnadieu, 2019) hence cannot be independently prescribed on tectonic timescale. For such timescale, $pCO_2$ is poorly constrained by proxy data and is among the least correlative climate indicators (Boucot & Gray, 2001; Eyles, 2008), yet it is conveniently invoked to cure many climate puzzles, including "faint-young-sun paradox" (FYSP, Crowley & North, 1991) and tropical glaciations (Pierrehumbert et al., 2011).




However uncertain, $p\mathrm{CO_2}$ reconstructed from paleosols falls far short in countering the dimmer sun (Sheldon et al., 2021) and even if it were sufficient, one still needs to explain why it would settle on this particular level that enables a habitable planet. In addition, $p\mathrm{CO_2}$ remains relatively steady through Neoproterozoic, displaying no wild swings envisaged by iceball earth (Sansjofre et al., 2011; Sheldon et al., 2021).

Another key internal variable lost amidst overprescribed $p\mathrm{CO_2}$ is the cloud: while three orders-of-magnitude greater $p\mathrm{CO_2}$ than the present is needed to counter the dimmer sun (Kasting, 1993), a mere 30% reduction in cloud is sufficient. Unlike $p\mathrm{CO_2}$, cloud has no proxy data, which moreover is not subjected to budget constraint as its generation/dissipation involves cloud physics of practically infinitive degrees of freedom. Clearly, unless the cloud is constrained by global thermodynamics, one does not have an explanation of Earth's temperature; for this reason, we shall retain only cloud to provide the requisite internal degree of freedom for our model closure. We should stress that absent a physical closure, no amount of outward sophistication of a climate model may advance its prognosis. Nor may such a model arbitrate our closure theory, which should be adjudicated only by its logical soundness and observation.

Since tropical glaciations are characterized by tectonic timescale, whatever their governing physics should also be operative for FYSP, we stipulate therefore that a valid thermodynamic closure must accommodate both phenomena and that they can be differentiated only by external condition. This uniformitarian view arguably precludes the runaway ice-albedo feedback since its triggering threshold is amply exceeded by the dimmer sun in Proterozoic, which however is largely free of glaciations. As for discernible external condition, we take note that tropical glaciations generally coincide with Precambrian supercontinents (Eyles, 2008; Williams et al., 2016) to posit that their concentration of the tropical landmass would block the brighter tropical sun to cause the required cooling.

To test this hypothesis in a most transparent manner, we construct a minimal box model, whose thermodynamic closure is first discussed in Sect. 2, to be followed in Sect. 3 by the derivation of glacial regimes. Given the observed paleogeography, we then apply the model prognosis in Sect. 4 to interpret glacio-epochs through Earth's history. We summarize the main findings in Sect. 5 to conclude the paper.

## 2 Thermodynamic Closure

Because of inherently turbulent nature of the planetary fluids, the climate is a macroscopic manifestation of a NT system hence likely subjected to maximum entropy production (MEP)—a generalized second law (Ozawa et al., 2003; Kleidon, 2009). Its predicate here on turbulence renders moot some its previous criticisms (Bruers, 2007) whose physical basis is further strengthened by its linkage to the fluctuation theorem (Dewar, 2005; Martyushev & Seleznev, 2006; Ou, 2018) as the latter is of considerable mathematical rigor and has been tested in the laboratory (Evans & Searle, 2002; Wang et al., 2002). For computational supports, MEP has emerged from general circulation models (Shimokawa & Ozawa, 2002; Kleidon et al., 2003) and eddy-resolving and direct numerical simulations of horizonal convection (Hogg & Gayen, 2020; Liu et al., 2022) have reproduced its signature feature (a mid-latitude front, Ou, 2006) absent from coarse-grained numerical models (Colin de





Verdière, 1988). In addition, MEP has replicated broad climate features and resolved some long-standing climate puzzles (Paltridge, 1975; Lorenz et al., 2001; Ou, 2023a).

With above supports, we have invoked MEP in our previous derivation of the generic climate state from first principles (Ou, 2001, 2006). The formulation is aided by the recognition that MEP is a selection rule hence can be applied hierarchically

(Lucarini et al., 2011)—first to the global-mean and then to the latitudinal fields, and since turbulence is internal to the fluids, MEP should apply to atmosphere and ocean individually. With the thermodynamic closure, certain derived climate properties may be regarded as known, which are first articulated below for the completeness of the present theory.

Through a global-mean model of aquaplanet, Ou (2001) demonstrates that clouds would self-adjust to stabilize the surface temperature constrained by intrinsic water properties. To illustrate this constraint, let us suppose that for a given forcing,

variable clouds allow a continuum surface temperature ($\bar{T}$) spanning the x-axis (linear in the corresponding blackbody radiance) of Fig. 1 and as an intrinsic water property, the saturation vapor pressure ($e$) rises increasingly sharply with warming surface (Clausius–Clapeyron relation). The accompanying long-wave (LW) absorption would increasingly differentiate surface ($q_{surf}$) and outgoing ($q_{OLR}$) LW fluxes (the greenhouse effect) to augment the LW cooling of the atmosphere (shaded, all radiative fluxes normalized by one-quarter of the present solar constant), which must be countered by increasing convective

flux from the surface ($q_c$). This convective flux is proportional to the product of turbulent wind ($u'$) and sea/air difference in the moist static energy, the latter roughly trending as the vapor pressure—except a finite cold asymptote because of the sensible heat. As such, the turbulent wind (thick line) first increases with the greenhouse warming, then decreases by the rising vapor pressure to render a local maximum (solid square). Since the turbulent wind varies with the frictional dissipation hence irreversible entropy production (Ou, 2001; Kleidon et al., 2014), MEP thus selects a temperature constrained by the intrinsic

water properties.

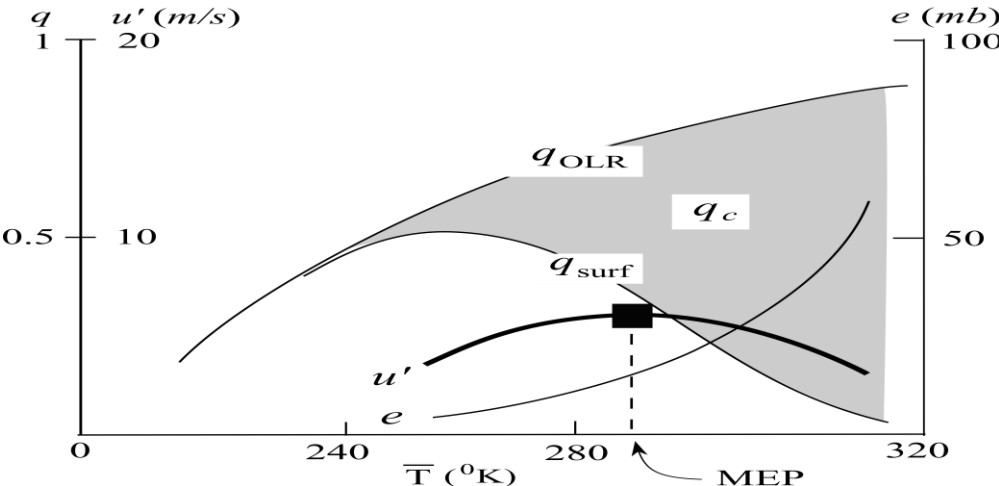

**Figure 1.** A schematic in which the saturation vapor pressure ($e$), LW fluxes ($q$'s, normalized by one-quarter of the present solar constant), convective flux ($q_c$, shaded), and turbulent wind ($u'$) are plotted against the surface temperature ($\bar{T}$, linear in the corresponding blackbody radiance). The maximum $u'$ (solid square) selects the MEP temperature.



Quantitatively, it is found that a 30% decrease of the solar insolation (about $100\ Wm^{-2}$) would be buffered by varying clouds to incur only $20\ Wm^{-2}$ reduction in the absorbed SW flux ($\bar{q}$) accompanied by $10\ ^0$C global cooling, resulting in a sensitivity of

$$s \equiv \delta\bar{T}/\delta\bar{q} \approx 0.5\ ^0C(Wm^{-2})^{-1}, \tag{1}$$

an often-quoted value (Pierrehumbert et al., 2011). It is seen therefore that so long as there is ocean (since at least Archaean,

Feulner, 2012), the surface temperature would be stabilized by MEP to within low-tens degrees above the freezing point, which thus may resolve the FYSP.

     With the global cloud known, Ou (2006, 2023b) then applies MEP to deduce its latitudinal tendency. To illustrate the basic physics, let us consider an ocean divided into warm/cold bands by a subtropical front, so its entropy production is simply the product of OHT and the differential temperature across the front. For a given thermal field, MEP then maximizes OHT,

which implies the same for the differential radiative forcing regulable by clouds. As such, the low cloud, which dominates the cloud albedo (Goldblatt & Zahnle, 2011), would be expelled to high latitudes, a deduction that is consistent with the observed vast stratus in high latitudes. With this tendency, we shall assume clouds as known, so the external forcing is the short-wave (SW) flux reaching the surface, which is subjected only to the land albedo.

     Applying MEP to the atmosphere and ocean successively, Ou (2006, 2018) links the differential sea-surface temperature

(SST, $T'$) to the differential forcing ($q'$) via

$$T' = q'/\alpha, \tag{2}$$

where $\alpha$ is the air-sea transfer coefficient augmented to include the latent heat. For an observational test, we set $\alpha = 15\ Wm^{-2}\ ^0C^{-1}$ (Ou, 2018, his appendix B) and differential forcing of $300\ Wm^{-2}$ (Peixoto & Oort, 1992, their figure 6.14), they would produce SST range of $20\ ^0$C, comparable to the observed one (Kucera et al., 2005). As an additional test, the MEP-

deduced MOC has a simple expression $\alpha A_{NA}(4\rho_o C_{p,o})^{-1}$ with $A_{NA}$ being the area of the North Atlantic cold band (poleward of $30^0\ N$), $\rho_o$, the ocean density, and $C_{p,o}$, its specific heat. Surprisingly, this MOC does not even depend on differential forcing, whose effect apparently has been neutralized by the thermal response (2). For a North Atlantic width of 6000 km, MOC would be 17 Sv, which is commensurate with the observed one (Macdonald, 1998). This constitutes a stringent test since it involves no free parameters—unlike previous comparisons involving tunable diapycnal diffusivity (Dalan et al., 2005), hence

provides a strong support of MEP.

     To recap, as a part of the present theory, prior derivations have determined the global temperature, its sensitivity to the global forcing (1), the linkage of the differential temperature and forcing (2), and latitudinal cloud—all have been tested against observation and can be regarded as known for the following derivations.

**3 Box model**

As a minimal representation of the observed ocean, we consider a box configuration as sketched in Fig. 2 where it is divided into tropical/polar bands of equal areas designated by numerals 1/2, and we assume for simplicity hemispheric symmetry so



only the hemispheric half is shown. As justified above, the external forcing is the SW flux reaching the surface ($q_i^*$), which, after reflection by land, is absorbed by the ocean ($q_i$) to differentiate the SST ($T_i$) and drive the OHT, the latter composed partly of random eddies. The total land area $A^*$ is known and its tropical portion $A$ is the independent variable whose effect on the climate is to be examined.

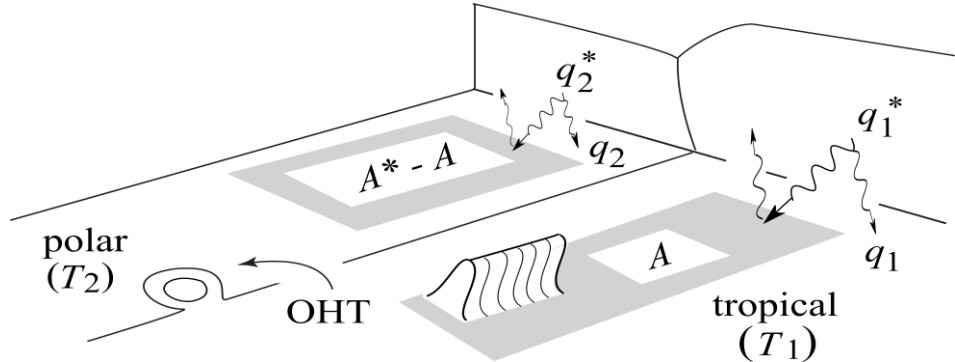

**Figure 2.** The model configuration in which the coupled ocean/atmosphere is divided into tropical/polar bands of equal areas designated by numerals 1/2. External forcing is the SW flux reaching the surface ($q_i^*$), which, after reflection by land, is absorbed by the ocean ($q_i$) to differentiate the SST ($T_i$) and drive the OHT composed partly of random eddies. The total land area $A^*$ is known and its tropical portion $A$ is the independent variable whose effect is to be examined.

Because of the thermal inertia of the ocean, we neglect its seasonality (Kucera et al., 2005) so the radiative forcing $q_i$ is the annual SW flux absorbed by the ocean. Although the SAT is seasonal, only the summer one is relevant in its control of the ablation hence the glacial margin. We assume the glacial margin to be zonal, so its marking temperature extends into the adjacent ocean, which moreover is assumed to be the same as the underlying SST in maintaining the radiative-convective equilibrium that defines the troposphere. In support of this assumption, the observed sea/air temperature difference in *summer* is no more than 2 $^0$C globally (Peixoto & Oort, 1992, their figure 10.7, lower panel). We assume continents to have mountain ranges reaching above the snowline, so there is always glacier to seed its advance to the sea-level when temperature falls.

### 3.1 Radiative forcing

We first determine the radiative forcing as a function of the tropical land area (all areas are expressed in fractions of the global surface). With land reflectance $r$, the radiative forcings are

$$q_1 = q_1^*(1 - 2rA), \tag{3}$$

$$q_2 = q_2^*[1 - 2r(A^* - A)], \tag{4}$$

where the numerical factor 2 stems from band areas being 1/2 of the global surface. The global radiative forcing is

$$\bar{q} \equiv (q_1 + q_2)/2, \tag{5}$$

which thus varies with the tropical land area as

$$\delta\bar{q} = -r\Delta q^* \delta A, \tag{6}$$

where





$$\Delta q^* \equiv q_1^* - q_2^* \tag{7}$$

is the differential SW flux reaching the surface, an external parameter. For a crude estimate of the Proterozoic forcing, we take

the current tropical/polar solar irradiance of $400/100\ Wm^{-2}$ (Peixoto & Oort, 1992, their figure 6.13), reduce it by 10%, and

apply cloud albedo of $0/0.5$ (based on its deduced tendency in Sect. 2) to yield the external forcing of $[\,q_1^*, q_2^*\,] =$

$[360, 45]\ Wm^{-2}$. With no land plants in the Proterozoic, the land reflectance is that of a desert set to $r = 0.5$ (Chandler &

Sohl, 2000), and for a total land area set to 0.3, we plot in Fig. 3a radiative forcings against the tropical land area $A$. It is seen

that as the land migrates into the tropics, it weakens the tropical forcing ($q_1$) more than it strengthens the polar forcing ($q_2$)

because of the brighter tropical sun, resulting in a net reduction of global forcing ($\bar{q}$) by $47\ Wm^{-2}$, which is quite substantial,

amounting to doubling the Proterozoic dimming of the sun from the present.

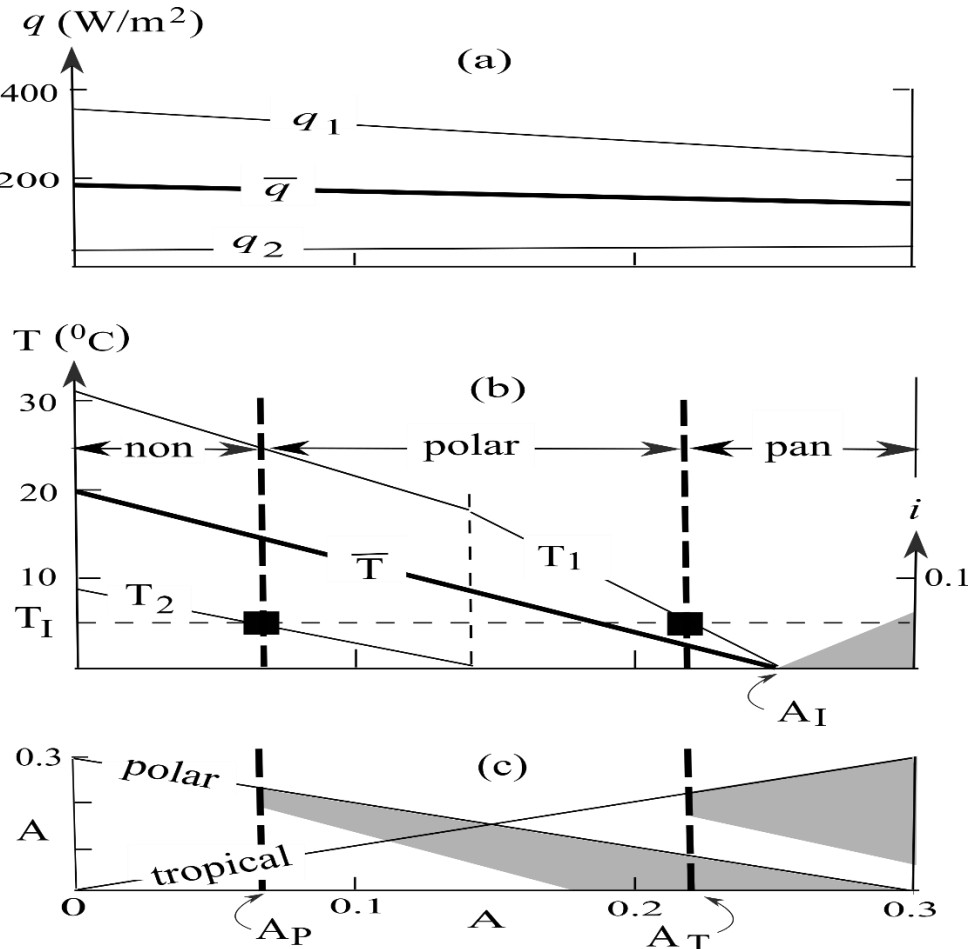

**Figure 3**. The radiative forcing (a), surface temperature (b) and land areas (c) plotted against the tropical land area $A$. Thick and thin lines
are global-means and tropical/polar values subscripted 1/2, respectively. The temperature $T_I$ marks the glacial margin whose intersections

with tropical/polar temperatures (solid rectangles) divide non/polar/pan-glacial regimes with $A_P/A_T$ being the thresholds for polar/tropical-
glaciation and $A_I$, the onset threshold for perennial sea ice (shaded). Shades in (c) signify the glaciated area.



### 3.2 Sea-surface temperature

Given the radiative forcing and sensitivity (1), global temperature is specified by its maximum set to 20 $^0$C (Scotese et al.,
2021) and plotted in the thick solid line in Fig. 3b. For the chosen parameter values, the global temperature would cool to the
freezing point with the tropical concentration of land but given the crudeness of the model and uncertain parameters, this
scenario should be regarded as merely plausible, which, as seen later, is not required for the tropical glaciation.

Subjected to (2), we plot tropical/polar temperatures in thin lines. Since $1/\alpha$ ($= 0.07$) is much smaller than $s$ ($= 0.5$), one
can see from (1) and (2) that both temperatures trend with the global cooling despite the increasing polar forcing. When the
polar temperature reaches the freezing point (thin vertical dashed line), the tropical temperature would drop more sharply with
$A$ until it is also cooled to the freezing point at $A_I$. As we shall see later, only when the tropical land area exceeds this threshold
($A > A_I$) would there be perennial sea-ice, as shaded.

### 3.3 Land glaciation

With mountain ranges providing seeding for the sea-level land ice, the latter would advance equatorward by accumulation
until halted by increasing ablation. Since ablation occurs only when the temperature is above the freezing point, so is the
marking temperature of the glacial margin to counter the finite yearly accumulation. Through crude ice dynamics and mass
balance, Ou (2023a) derives an expression for this marking temperature whose prognosed value $T_I \approx 5$ $^0$C is consistent with
the present Greenland ice sheet (Oerlemans, 1991). This marking temperature is the horizonal dashed line of Fig. 3b whose
intersections with tropical/polar temperatures (solid rectangles) then divide non/polar/pan-glacial regimes (thick vertical
dashed lines) with $A_P/A_T$ denoting the thresholds when ice would advance into the polar/tropical land, respectively.

For a box model, one obviously may not discern the glacial margin within each band or its longitudinal excursion on land,
the above deduction nonetheless allows one to assess broad glacial extent, and to aid the visualization, we show in Fig. 3c
tropical/polar land areas and their glaciated portions signified by shades. Increasing tropical land naturally shrinks the polar
land, but the attendant cooling would expand the polar glaciation until it is limited by the available polar land, and when $A_T$ is
exceeded, there would be a strong glacial expansion into the tropical band—even when the polar glaciated surface shrinks to
zero, as is the observed case (Evans & Raub, 2011). With vagaries of the tectonics that control the tropical land area $A$ (that is,
moving along the $x$-axis), one thus expects glacial advance/retreat on tectonic timescale, as observed (Rieu et al., 2007; Allen
& Etienne, 2008), so should be the onset/termination of the tropical glaciation when $A_T$ is crossed—not the millennial
timescale governing the ice mass balance or the ice-albedo feedback (Ou, 2023a; Hyde et al., 2000).

Figure 3c points to a wide-spread misconception that increasing polar land always favors glaciation (Crowley & Baum,
1993): As readily seen from the figure that this holds only if the polar land is already saturated by ice—for otherwise the
accompanying warming would in fact shrink the glaciated surface to possibly vault it into the non-glacial regime, the latter
thus may resolve the seeming puzzle of warm Cambrian despite polar positioning of Gondwana (Evans, 2003; Li et al., 2013).
Since pan-glacial regime hinges on global cooling, tropical glaciations should be globally synchronous, as is the observed case





(Rooney et al., 2015; Hoffman et al., 2017), which incidentally invalidates the zipper-rift hypothesis of regional glaciation
(Eyles & Januszczak, 2004). And then since the polar land is always glaciated in pan-glacial regime, it agrees with observation
(Evans, 2003, his figure 7) but dispels the high-obliquity conjecture of Williams et al. (1998).

### 3.4 Perennial sea ice

The sea ice may form only when the polar water has cooled to the freezing point, and with the differential temperature fixed
as such, MEP implies a maximized OHT. This leads immediately to vanishing perennial ice since any such ice would curb the
ocean cooling hence OHT, contravening its maximization. One should note however that MEP operates on millennial timescale
(Ou, 2018) so it does not preclude sea ice formation over shorter timescale, including the seasonal one. In fact, with the SST
hovering around the freezing point, there is necessarily extensive winter ice, as seen during the last ice age, but tellingly the
North Atlantic remains largely open in summer (de Vernal et al., 2005), a peculiarity that may be readily explained by MEP.
Extending the above argument, we see that only when the tropical ocean is also cooled to the freezing point with the shutdown
of OHT could there be perennial sea ice—*even* over the polar ocean. At this juncture $A_I$, the cooling flux at the ocean surface
$\bar{q}_o$ has reached its lower limit given by the radiative forcing $\bar{q}_I$,

$$\bar{q}_o = \bar{q}_I, \tag{8}$$

so any further reduction of the radiative forcing (that is, $\bar{q} < \bar{q}_I$) must induce sea-ice cover ($i$) to maintain the ocean heat
balance

$$\bar{q}_o (1 - i) = \bar{q} - i q_2^*, \tag{9}$$

where the last term represents the positive feedback between sea ice and the radiative forcing (assuming sea ice remains in the
polar band, to be checked later). Substituting (8) into (9), we obtain

$$i = \frac{1 - \bar{q}/\bar{q}_I}{1 - q_2^*/\bar{q}_I}, \tag{10}$$

the sea ice thus is fully specified by radiative forcing, as shaded in Fig. 3b. It increases linearly with decreasing $\bar{q}$ and attains
a maximum of about 6% of the global surface, a deduction that is consistent with the observation that there is little sea ice for
all glacio-epochs (Eyles, 2008). Equation (10) also shows the importance of positive ice-albedo feedback without which the
sea ice cover would be halved.

It is seen therefore that even when the global ocean is cooled to the freezing point, the perennial ice remains confined to
high latitudes, so despite crudeness of our box model, the tropical ocean for all practical purposes should remain open to
maintain active hydrological cycle, as observed but contrary to iceball earth.

### 4 Glacio-epochs chronology

We have identified non/polar/pan-glacial regimes controlled by the tropical appropriation of land and based on the observed
paleogeography (Eyles, 2008; Bradley, 2011; Li et al., 2013; Nance et al., 2014), we offer the following interpretation of the
glacio-epochs aided by the numbered stages shown in Fig. 4 (time moves to the left).



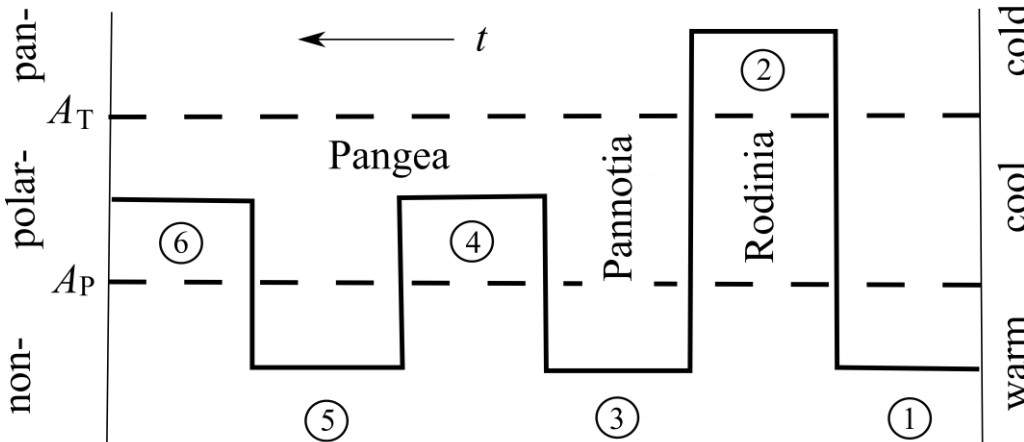


**Figure 4.** The model interpretation of glacio-epochs based on observed paleogeography (time moves to left), aided by numbered stages. Dashed lines are tropical-land thresholds that divide non/polar/pan-glacial regimes.

Stage 1 represents the bulk of Proterozoic when there are no supercontinents to concentrate the tropical land and, with the fainter sun being moderated by less cloud (Sect. 2), the climate is too warm for glaciation. With our hypothesis of glacio-
epochs in equilibrium with paleogeography, there is thus no significant puzzle for a billion-year long glacial lull (Pierrehumbert et al., 2011).

Stage 2 represents the tropical tenure of supercontinents, which would block the brighter tropical sun to cause the tropical glaciation. This pan-glacial state seems to occur to all Precambrian supercontinents (named glaciations indicated)—definitely to Ur (Pongola), Kenorland (Huronian), and Rodinia (Sturtian, Marinoan, Gaskiers), and possibly to Columbia (nameless)
(Eyles, 2008; Young, 2013; Williams et al., 2016). Although glacial deposits are mostly found in rift basins, it could reflect preservation bias (Eyles, 2008) since weathering is also elevated during assembly of supercontinents (Maruyama & Santosh, 2008; Macdonald et al., 2019) and indeed Huronian glacial deposits precede the breakup of Kenorland (Young, 2013). Bypassing the questionable role of $p$CO$_2$ (Kerrick & Caldeira, 1999, see also Sect. 1) and uncertain paleogeography, we can only ascribe pan-glacial regime to the tropical cluster of landmasses not their detailed evolution. The vagaries of such evolution
on tectonic timescale may accommodate disparate durations of Sturtian/Marinoan/Gaskiers (50/5/1 million years) (Li et al., 2013; Stern & Miller, 2021), which are difficult to bridge by iceball earth because of the minimum time needed to build up $p$CO$_2$ for deglaciation (Hoffman et al., 1998; Goddéris et al., 2021).

Stage 3 marks the breakup of Rodinia, which reassembles as Pannotia near the south pole during early Paleozoic (Li et al., 2013; Murphy et al., 2021). The lack of tropical land has propelled the climate to non-glacial regime as seen in the warm
Cambrian (Hearing et al., 2018, see also Sect. 3.3). The warmth is temporarily reversed in late Ordovician when Laurentia and Baltica rift away from Gondwana and enter the tropics (Marcilly et al., 2022), causing the Andean-Saharan glaciation, which thus need not involve volcanic eruption or bolide collision (Scotese et al., 2021).

Stage 4 represents the assembly of Pangea when Laurentia reattaches Gondwana to extend from pole-to-pole (Nance et al., 2014). Although there is more tropical land than Pannotia, the sizable polar land hence the associated warmth only allows



polar-glaciation (Karoo) lasting about 100 million years (Eyles, 2008). Based on our model, it is thus the smaller tropical land of Pangea in comparison with Precambrian supercontinents that distinguishes Phanerozoic and Precambrian glacio-epochs (Evans, 2000), which otherwise are subjected to the same physics in adherence to the uniformitarianism (Etienne et al., 2007), again there is no need to invoke extraneous and unsupported physics of iceball earth.

Stage 5 represents the breakup of Pangea that propels land into high northern latitudes (Eyles, 2008), the ensuing warmth

may account for the non-glacial interval spanning Mesozoic and early Cenozoic. Together with Stage 3, they correspond to the classical double humps (Fischer, 1982). This warming reverses in Eocene when climate begins to cool (Eyles, 2008), which has been attributed to the uplift of Himalayas and enhanced weathering (Raymo & Ruddiman, 1992), whose efficacy however has been questioned (Kerrick & Caldeira, 1999). Adhering to the paleogeographic control, it is also possible to ascribe this cooling to the polar centering of Antarctica accompanied by opening of the Drake Passage (Eyles, 2008), which would have

deprived the Antarctic interior from the ocean heat, causing it to icing over accompanied by global cooling, a conjecture that is examined in a forthcoming paper.

Stage 6 represents the current polar-glacial regime (Quaternary) (Young, 2013). The contiguous northern land extending into the subtropics allows glacial advance/retreat in response to the orbital forcing (Milankovitch, 1941). Applying MEP in a box model, Ou (2023a) shows that Cenozoic cooling may account for the mid-Pleistocene transition from obliquity- to

eccentricity-dominated glacial cycles, which thus can be integrated into the deep-time framework presented here. The orbital-period glacial cycles however should be distinguished from aperiodic glacio-epochs of tectonic timescale (Allen & Etienne, 2008).

**5 Conclusion**

An extreme scenario was previously proposed to explain Cryogenian tropical glaciations, which invokes runaway ice-albedo

feedback leading to an iceball earth. In addition to its conflict with sedimentary data showing an open ocean, we point out flawed physics of the runaway scenario and deficient clousure of numerical models in addressing the problem. As an alternative, we advance the uniformitarian view that the thermodynamic closure of the system must accommodate both FYSP and tropical glaciation, and that the two should be distinguished only by external conditions.

For the thermodynamic closure, we posit that, because of the inherently turbulent nature of the planetary fluids, the generic

climate state is a macroscopic manifestation of a NT system hence characterized by maximum entropy production (MEP), a generalized second law. And for discernible external conditions, we posit that Precambrian supercontinents may block the brighter tropical sun to cause tropical glaciation. To test this hypothesis in a most transparent manner, we formulate a minimal tropical/polar box model to examine the effect of differential land partition on the climate and glacial regimes. As a part of the present theory, prior applications of MEP by this author have determined certain thermal properties and together with new

findings, they are summarized below.



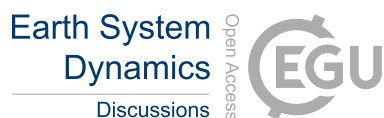

- Applying MEP in a global-mean aquaplanet model, Ou (2001) shows that global cloud would self-adjust to render habitable temperature constrained by intrinsic water properties (Clausius–Clapeyron relation), thereby resolving the FYSP. The closure allows the determination of the global temperature and its sensitivity to the radiative forcing.

- Applying MEP to a latitudinal model, Ou (2006) shows that low clouds, which dominate the cloud albedo, are expelled to high latitudes, thereby maximizing the differential forcing, the latter in turn specifies the differential temperature.

- Applying MEP and foregoing constraints in a box model, we produce a regime diagram showing cooling tropical/polar temperatures when increasing tropical land would block the brighter tropical sun. Even when the global ocean is cooled to the freezing point, the perennial sea ice remains confined to high latitudes, thwarting therefore the runaway ice-albedo feedback.

- The glacial margin is marked by above-freezing temperature to counter the yearly accumulation, whose crossing by tropical/polar temperatures would divide non/polar/pan-glacial regimes, the last being characterized by tropical glaciation abutting an open ocean.

- Given the observed paleogeography, the model may provide a unified account of glacio-epochs through Earth's history. Precambrian and Phanerozoic glacio-epochs differ only by tropical appropriation of the land in its blockage of the brighter tropical sun.

In conclusion, through a minimal box model subjected to MEP, our theory provides a unified account of FYSP, Precambrian tropical glaciations, and glacio-epochs through Earth's history—all are differentiated by paleogeography varying on tectonic timescale, in support of the uniformitarian principle.

**Data availability:** No proprietary data are used.

**Competing interests:** This author declares no conflicting interest relevant to this study.

**Acknowledgments:** None

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
