# Peer review of "Linkage of tropical glaciation to supercontinents: a thermodynamic closure model"

_Earth System Dynamics, 2023_

## Author Comment (AC1)

First of all I want to clarify that I was not familiar with the concept of maximum entropy production (MEP) before reviewing this manuscript and therefore will not attempt to judge, whether the basic and additional assumptions of this concept are valid here. Thus, I put my focus on whether I can follow the arguments presented here and how to align the conclusions with previous studies.

I first answer some general questions regarding the manuscript. Then I provide a general comment, as well as some specific further remarks and references supporting my comments.

- *Thank you for the unusually insightful and detailed comments, which have further crystallized my own thinking. Since you have waived anonymity, I shall acknowledge your comments explicitly in the revised paper.*

- *MEP is indeed unfamiliar to most modelers, but its predicate of an eddying ocean is difficult to bypass unless models have resolved eddies—significant carriers of OHT. To aid the understanding of MEP, I shall add the differential heat equation in Section 2 to derive the differential temperature and MOC (meridional overturning circulation) and show that both contain no free parameters hence can be veritably tested against observation. This equation also aids the later expanded discussion of the ice-albedo feedback (Section 3.4).*

1. **Does the paper address relevant scientific questions within the scope of ESD?**

Yes, the manuscript discusses implications of the postulated basic thermodynamic principle of maximum entropy production on the interactions of continental configuration, ocean heat transport and land/sea ice extent.

2. **Does the paper present novel concepts, ideas, tools, or data?**

I would prefer leaving the judgement of this to the editor and other reviewers. The presented core idea is relating the major cycles of global glaciation on Earth to the continental configuration based on the concept of maximum entropy production using a highly idealized box model.

3. **Are substantial conclusions reached?**

Yes, the main conclusions reached is the claim that the major cycles of global glaciation (none, polar, and pan-glaciation) are determined by the continental configuration only and the sea-ice albedo feedback plays a negligible role for pan-glacial states during Earth's history.

4. **re the scientific methods and assumptions valid and clearly outlined?**

I have problems of relating my understanding of the ice-albedo feedback with the applied model

One assumption that from my point of view is key to the importance of the presented results is the treatment of the ice-albedo feedback in the applied model. This is quite unclear to me.

An explicit assumption regarding ocean albedo and how it changes with the formation of sea ice as well as the albedo of snow/ice covered land is not made. However, in line 246 it is mentioned that there is an ice-albedo feedback in the model. How is it incorporated?

- *I plan to expand the discussion of ice-albedo feedback by way of bifurcation diagram, which progresses from iceball earth to tropical waterbelt (Braun 2022a) to divergent land-ice line to MEP. I will show that differing land/sea-ice lines are sufficient to render tropical glaciation*

*abutting open ocean, thus forgoing the need of waterbelt, and the sea-ice line moreover would be expelled to the pole by MEP, thus thwarting the runaway ice-albedo.*

- *With the added differential heat equation, it's readily seen that ocean would be open so long as there is differential temperature to institute OHT maximized by MEP. Only when the tropical ocean is also cooled to the freezing point in shutting down of OHT could there be runaway ice-albedo to produce the iceball. I shall further argue however that such iceball is untenable when MEP is applied to the global-mean field (Section 2).*

- *It is easy to argue that the land-ice line is marked by above-freezing temperature since you need finite ablation to counter the accumulation. I shall add discussion about how this marking temperature can be derived (Ou 2023) and that its deduced value (~ 5 $^0C$) is commensurate with the current observation of the Greenland ice sheet.*

Another aspect that from my point of view may have a strong impact on the sea ice extent, but is not considered here, is the potential formation of high latitude sea glaciers that may be ~1km thick and thus cause gravitational spread of sea ice towards the equator (see Fig. 16 in Hoffman et al., 2017). Please clarify, or justify why the inclusion of this effect may not be necessary.

- *Sea glacier is preconditioned on sea ice formation, both are products of models and remain to be validated by observation, as noted by Eyles (2008)"no sedimentary evidence of sea ice for all glacio-epochs, contrary to models".*

Lines 228 to 230: This content relates to my questions regarding sea ice. I do not understand the following formulation:

"and with the differential temperature fixed as such, MEP implies a maximized OHT. This leads immediately to vanishing perennial ice since any such ice would curb the ocean cooling hence OHT, contravening its maximization"

- *With the added differential heat equation, I shall expand the relevant discussion, as seen below.*

Specific points, I do not understand, are:

1) To what (value/state) is the differential temperature fixed, and why does MEP then imply maximized OHT?

- *A freezing-point cold box uniquely specifies the warm-box temperature (given the global-mean temperature, see Fig. 3) hence the differential temperature. Since entropy production is the product of OHT and the differential temperature, MEP would maximize the OHT.*

2) Is atmospheric heat transport also considered here? If not, why?

- *This will be included in the added discussion of Section 2. Specifically, MEP of atmosphere would link differential SAT to SST and yields equipartition of ocean/atmosphere heat transport at mid-latitudes, the latter being broadly consistent with observation.*

3) Why do sea ice formation and maximized OHT exclude each other?

- *Sea ice would block the ocean cooling to reduce OHT, contravening MEP. On the other hand, MEP operates on millennial timescale (Ou 2018) hence does not preclude seasonal ice. This is*

*the reason that although the subpolar water is covered by winter ice during the last ice age, it remains open in summer (de Vernal et al. 2005). This observation incidentally conflicts with the ice line depicted in prevailing bifurcation diagrams.*

Besides that, mostly, the assumptions are clearly outlined. Some of the assumptions require further clarification; see list below.

Line 164: Assumption on orography to produce glacier flow towards sea. In this context a discussion of Walsh et al., 2019 might be beneficial.

- *This paragraph is confusing and will be rewritten. Thanks for pointing out Walsh et al. (2019), which is indeed relevant. In particular, the presence of highland is critical in seeding the sea-level ice sheet, which is not sufficiently emphasized.*

line 181: The assumption on cloud albedo seems very strong, given the zonal-mean cloud albedo of present-day climate; cf e.g. Södergren and McDonald, 2022; Fig. 1 c). I suggest to further justify this assumption and include a sensitivity test of the derived results regarding this assumption. E.g., how would results look like for polar/tropical cloud albedo of 0.4/0.2 ?

- *If one sets cloud reflectance of 0.7 (Hartmann 2015, his Table 3.2) and 70% cloud cover in high-latitudes (Michibata et al. 2019, their Fig. 1), the cloud albedo would be 0.5, which is commensurate with observation (Donohoe and Battisti 2011, their Fig. 3; Södergren and McDonald 2022, their Fig. 1c). I will add this discussion in the revision.*

- *I will discuss qualitative dependence and sensitivity of the model solution. I shall also stress that given the crude model and uncertain parameters, the model results should be interpreted as qualitative and plausible.*

line 182: land reflectance. I suggest to also include a sensitivity test here.

- *I plan to reduce land reflectance from 0.5 to 0.3 to conform more closely to observation (Hartmann 2015, his Table 4.2 for desert). A sensitivity statement will be included, as seen above.*

5. **Are the results sufficient to support the interpretations and conclusions?**

Besides my points lined out above (4.): If accepting the principle of MEP and the assumptions made throughout the manuscript, the presented theory seems to support the interpretations and conclusions.

- *As I stated earlier, I will expand the discussion of MEP.*

6. **Is the description of experiments and calculations sufficiently complete and precise to allow their reproduction by fellow scientists (traceability of results)?**

Yes, as far as I see the equations provided throughout the manuscript allow the calculation of the qualitative results shown in the figures.

I have two points, though:

L173: Although it is explained, I do not understand why the factor 2 needs to be taken into account here.

- *I will rephrase the sentence: Since A is the fractional area of the global surface, A/(1/2)=2A is its fraction of the cold box hence an albedo of 2Ar. I might have to add Ou (2023b) about hemispheric symmetry of the polar albedo.*

Regarding Fig 3c: Why is there unglaciated polar land for T2 = 0°C ?

- *Keen observation, I shall amend the drawing.*

7. **Do the authors give proper credit to related work and clearly indicate their own new/original contribution?**

Yes.

8. **Does the title clearly reflect the contents of the paper?**

Yes.

9. **Does the abstract provide a concise and complete summary?**

Yes.

10. **Is the overall presentation well structured and clear?**

The structure of the manuscript is clear. However I have problems in relating sections 2 and 3.

- *Valid point, I shall strengthen the logical progression from Section 2 to 3.*

E.g., in section 2 the author refers to MEP based on turbulent wind, whereas in section 3, the author discusses MEP in the context of ocean heat transport.

- *MEP is a selection rule hence can be applied hierarchically first to the global-mean (Ou 2001) and then the differential fields. For the global mean, entropy production is roughly the surface dissipation hence varies as turbulent wind. For the differential field, on the other hand, the entropy production is dominated by latitudinal variation of temperature—not the turbulent wind, which would translate to the product of OHT with differential temperature. I will add this point in the revision.*

11. **Is the language fluent and precise?**

In my perception the language used throughout the article hinders to access the content of the article. This is mainly due to some of the used vocabulary and sentence structure. To some extent the language even appears offensive to me (e.g. the usage of the word "flawed").

- *I will moderate my critique.*

12. **Are mathematical formulae, symbols, abbreviations, and units correctly defined and used?**

In Fig. 3 b) the meaning of "I" labelling the right hand side axis is not mentioned in the figure caption.

- *The ice extent is wrong, it should cover the globe ocean (see Response 2 to Comment 4).*

It is not clear to me, whether A and A* refer to land area or land area fraction.

- *Besides stating at the beginning of Section 3.1 that "all areas are expressed in fractions of the global surface", I shall add it to the caption of Fig. 3.*

13. **Should any parts of the paper (text, formulae, figures, tables) be clarified, reduced, combined, or eliminated?**

As mentioned in 10. I would appreciate more guidance on relating sections 2 and 3.

- *See Response 1 to Comment 10.*

14. **Are the number and quality of references appropriate?**

Yes

15. **Is the amount and quality of supplementary material appropriate?**

There is no supplementary material. I would appreciate if a sensitivity study of relevant parameters would be added here (see point 4 for suggestions).

- *See Response 11 to Comment 4.*

**General comment**

From my point of view more complex approaches of modeling climate states of Earth's past based on dynamic three-dimensional equations serve to focus on single aspects of Earth system dynamics. Yet, I see the point of the author, that these approaches are highly idealized and thus may overlook the interaction of fundamental processes. I do not see a strong conflict between the approach based on MEP presented here and more complex modeling approaches of pan-glacial states on Earth. Thus, I would appreciate and recommend to include a discussion of how the presented main conclusion in this manuscript relates to more complex modeling approaches lined out below. The points below are partly taken from Braun, 2022 chapter 2.3. Cryogenian waterbelt scenarios; see there for further details.

- *There should be no difference between MEP and primitive-equation models if the latter have resolved eddies. The nonclosure of the latter lies in sub-grid parameterization containing free parameters, such as diapycnal diffusivity, on which the MOC depends sensitively hence is finely tuned to yield the current climate. MEP on the hand contains no free parameters (should be distinguished from uncertain but physical quantities) hence is potentially prognostic. Paradoxically, since MEP-deduced MOC depends only on air/sea exchange coefficient for a temperate state, climate models of tuned MOC may retain its prognostic utility of a changed climate, the difficulty however amplifies for the glacial state when MOC is no longer fixed but maximized. Resolving eddies requires grid spacing of 0.3 degree or less, which obviously poses a significant challenge, but there is a possible phenomenological approach in bypassing this*

*requirement, as discussed in Ou (2023a, his Section 5). I plan to include the above point in my revision.*

1) Hyde et al., 2000 found tropical land ice with a sea ice edge located at 25° lat and a continuous transition between this climate state and climate states with less ice cover, i.e. without an ice-albedo feedback. They applied "GCM GENESIS 2 coupled to a thermodynamic mixed-layer ocean and thermodynamic sea ice, neglecting ocean and sea-ice dynamics".

- *They used ice-sheet model to produce ice-covered land, which is then coupled to the ocean. Since the sea ice has not reached the bifurcation point, it remains regulable by forcing. Although their mixed-layer ocean provides only minor heating, it is nonetheless sufficient to keep the sea ice at bay given their forcing. The paper underscores the importance of ice-sheet dynamics, which generally would produce more expansive land- than sea-ice.*

2) Based on simulations with the GCM FOAM, Poulsen et al. (2001) "found iceball states with mixed-layer ocean but not with a fully coupled ocean", i.e. in their model "ocean heat transport counteracts the expansion of sea ice towards the equator".

- *They show that ocean dynamics would strongly augment OHT of a mixed-layer ocean in deterring the iceball earth. But since their grid spacing is greater than 1 degree hence does not resolve eddies, we argue that it has not fully captured the OHT effect.*

3) Based on a "coupled ocean–atmosphere model of intermediate complexity (CLIMBER-2)", "Donnadieu et al. (2004b) considered the major stabilizing effect on the simulated waterbelt climate to be meridional atmospheric heat transport via the Hadley circulation."

- *Their ocean grids are 2.5 degrees wide, too coarse to resolve eddies. I also have reservation about prescribing pCO2 to probe the climate response (see Introduction).*

4) Rose (2015) found that "ocean heat convergence at the ice edge driven by a feedback between ice extent, wind stress and ocean circulation was found to stabilize the waterbelt state".

- *Many mechanisms have been proposed in stabilizing the waterbelt, which is not a robust feature (Braun et al. 2022b). In any event, as I discussed in Responses 1 and 2 to Comment 4, one need not invoke tropical waterbelt to maintain an open ocean.*

**Further remarks**

l30: Please explain abbreviation pCO2 l55: Please clarify the statement regarding the unbalanced initial state.

- *I will add its descriptor: the partial pressure of atmospheric CO2.*

- *The sudden reduction of solar forcing would cause sea ice to grow to the bifurcation point before OHT operating on decadal timescale has time to act, perhaps the reason they see no difference if OHT is shut off. I will expand the discussion of this point.*

L78: What does external conditions explicitly refer to?

- *The external perturbation of the climate is the tectonics, which is the changing paleogeography in our case. I will add this statement.*

L105: Please explain the expression "linear in the corresponding blackbody 105 radiance".

- *I should just state "x-axis is the blackbody radiance of the surface". In any event, I will remove this figure since without the full closure considered in Ou (2001), the figure raises more questions than it answers.*

L162 to 165: I find it hard to follow and understand the argument about the marking temperature (here and throughout the text)

- *I will expand the discussion of the glacial marking temperature (perhaps change to land-ice line temperature?) and refer to Ou (2023a) for details. It involves ice dynamics and mass balance, rendering this temperature an intrinsic property of the ice sheet, and the deduced temperature of about $5^0C$ is consistent with the present Greenland ice sheet.*

l225: High obliquity hypothesis: The box model assumes a specific zonal distribution of radiative forcing to polar/tropical boxes. The high obliquity hypothesis is based on the recognition that zonal distribution of radiative forcing changes. Thus, the distribution of radiative forcing to polar and tropical regions in the box model would need to be adjusted for application to a high obliquity state. I suggest to discuss this or apply the box model with modified radiative forcing to further investigate this statement.

- *The high obliquity hypothesis assumes reversal of differential heating. There is no evidence of such orbital or glaciation reversal.*

L314, 315: See my point regarding cloud albedo in present-day climate above.

- *See Response 10 to Comment 4.*

**References**

Hoffman, Paul F., Dorian S. Abbot, Yosef Ashkenazy, Douglas I. Benn, Jochen J. Brocks, Phoebe A. Cohen, Grant M. Cox, et al. "Snowball Earth Climate Dynamics and Cryogenian Geology-Geobiology." *Science Advances* 3, no. 11 (2017). https://doi.org/10.1126/sciadv.1600983.

Walsh, Amber, Thomas Ball, and David M Schultz. "Extreme Sensitivity in Snowball Earth Formation to Mountains on PaleoProterozoic Supercontinents." *Scientific Reports* 9, no. 1 (2019): 1–7. https://doi.org/10.1038/s41598-019-38839-6.

Södergren, A. H., and A. J. McDonald. "Quantifying the Role of Atmospheric and Surface Albedo on Polar Amplification Using Satellite Observations and CMIP6 Model Output." *Journal of Geophysical Research: Atmospheres* 127, no. 12 (2022): e2021JD035058. https://doi.org/10.1029/2021JD035058.

Braun, dissertation 2022: https://publikationen.bibliothek.kit.edu/1000150229

Hyde, William T, Thomas J Crowley, Steven K Baum, and W Richard Peltier. "Neoproterozoic 'Snowball Earth' Simulations with a Coupled Climate/Ice-Sheet Model." *Nature* 405, no. 6785 (2000): 425–29. https://doi.org/10.1038/35013005.

Poulsen, Christopher J., Raymond T. Pierrehumbert, and Robert L. Jacob. "Impact of Ocean Dynamics on the Simulation of the Neoproterozoic 'Snowball Earth.'" *Geophysical Research Letters* 28, no. 8 (2001): 1575–78. https://doi.org/10.1029/2000GL012058.

Donnadieu, Yannick, Yves Goddéris, Gilles Ramstein, Anne Nédélec, and Joseph Meert. "A 'Snowball Earth' Climate Triggered by Continental Break-up through Changes in Runoff." *Nature* 428, no. 6980 (2004): 303–6.

Rose, Brian E. J. "Stable 'Waterbelt' Climates Controlled by Tropical Ocean Heat Transport: A Nonlinear Coupled Climate Mechanism of Relevance to Snowball Earth." *Journal of Geophysical Research: Atmospheres* 120, no. 4 (2015): 1404–23. https://doi.org/10.1002/2014JD022659.

*Cited references:*

*Braun, dissertation 2022a: https://publikationen.bibliothek.kit.edu/1000150229*

*Braun C, Hörner J, Voigt A, Pinto JG (2022b) Ice-free tropical waterbelt for Snowball Earth events questioned by uncertain clouds. Nat Geosci (6):489-93 https://doi.org/10.1038/s41561-022-00950-1*

*de Vernal A, Eynaud F, Henry M, Hillaire-Marcel C, Londeix L, Mangin S, Matthiessen J, Mar-ret F, Radi T, Rochon A, Solignac S, Turon JL (2005) Reconstruction of sea surface conditions at middle to high latitudes of the Northern Hemisphere during the Last Glacial Maxi-mum (LGM) based on dinoflagellate cyst assemblages. Quat Sci Rev (24):897–924 https://doi.org/10.1016/j.quascirev.2004.06.014*

*Donohoe A, Battisti DS (2011) Atmospheric and surface contributions to planetary albedo. J Clim 24:4402-18 https://doi.org/10.1175/2011jcli3946.1*

*Eyles N (2008) Glacio-epochs and the supercontinent cycle after ~ 3.0 Ga: tectonic boundary conditions for glaciation. Palaeogeogr Palaeoclimatol Palaeoecol 258(1-2):89-129 https://doi.org/10.1016/j.palaeo.2007.09.021*

*Hartmann DL (2015) Global physical climatology. Academic Press, New York, 411 pp*

*Klein SA, Hartmann DL (1993) The seasonal cycle of low stratiform clouds. J Clim 6(8):1587-606 https://doi.org/10.1175/1520-0442(1993)006<1587:tscols>2.0.co;2*

*Michibata T, Suzuki K, Sekiguchi M, Takemura T. Prognostic precipitation in the MIROC6-SPRINTARS GCM: Description and evaluation against satellite observations. Journal of Advances in Modeling Earth Systems. 2019 Mar;11(3):839-60.*

*Ou HW (2018) Thermohaline circulation: a missing equation and its climate change implications. Clim Dyn 50:641-53 https://doi.org/10.1007/s00382-017-3632-y*

*Ou HW (2023a) A theory of orbital-forced glacial cycles: resolving Pleistocene puzzles. J Mar Sci Eng 11(3):564 https://doi.org/10.3390/jmse11030564*

*Ou HW (2023b) Hemispheric symmetry of planetary albedo: a corollary of nonequilibrium thermodynamics. Atmosphere 14:1243 https://doi.org/10.3390/atmos14081243*

---

## Author Comment (AC2)

Title: Linkage of tropical glaciation to supercontinents: a thermodynamic closure model

Author: Hsien-Wang Ou

In this paper, the author uses a simplified two-box model of the Earth and investigates how the tropical and polar temperature distribution changes with the change in the area of tropical land. The main finding is that, as the tropical land area increases, solar radiation is reflected effectively, and both tropical and polar temperatures tend to decrease towards the ice-forming temperature, although the land area in the polar region decreases. This tendency may explain tropical glaciations due to the appearance of Precambrian supercontinents, as well as some subsequent glacial epochs triggered by continental drifts. This reviewer finds these results basically interesting. However, the model assumes intricate relationships between radiations and temperatures that are not explained explicitly in this paper. Also, some of the model parameters and assumptions are stated so vaguely that readers of this paper cannot verify the validity of the results obtained. This reviewer therefore recommends major revisions of this paper regarding the comments below.

- *I want to thank you for the highly substantive comments, which point out many shortfalls of the paper. Together with the other reviewer's comments, I plan to thoroughly revamp the paper, as contained in my response given below in the italics.*

- *I will add the differential heat equation and show how MEP would link temperature to the forcing. I will expand discussions of parameters and assumptions as detailed below.*

Major comments:

1. Equation (1) and the sensitivity of s = 0.5 °C/(W/m$^2$).

This sensitivity (0.5) is used to estimate the dependence of the global-mean temperature on the tropical land area A in Fig. 3b, and therefore plays a significant role. This value, however, is quite large compared to the well-known theoretical value calculated from the Stefan-Boltzmann law: F = $\sigma T^4$, from which one can derive dT/dF = $1/(4 \sigma T^3) \approx 0.2$ for T $\approx$ 280 K.  Although the used value is more than twice as large as the well-known theoretical value, the rationale for this is not clearly explained in the text or the cited reference, making the estimate somewhat doubtful. Please provide a valid reason for using this value.

- *"Global" sensitivity is the change of the global surface temperature given that of the global forcing, the latter being the absorbed SW flux or the outgoing LW flux as you have intended. This flux however is quite smaller than the blackbody radiance because of the greenhouse effect. If one uses a LW window of 0.3, it would augment your sensitivity estimate three-fold to 0.6, which is still near the low end of its calculated range [0.5,1.5] (Manabe and Wetherald 1967, their Table 5; Bjordal et al. 2020), so to conform to these calculations, I plan to set the global sensitivity to unity. I will add above discussion in the revision.*

2. Equation (2) and the air-sea transfer coefficient of $\alpha$ = 15 (W/m$^2$/°C).

This coefficient implies the reciprocal of the temperature sensitivity, and is used to estimate the tropical and polar temperatures from the deviation of solar radiation from the average. The corresponding sensitivity, $1/\alpha \approx 0.07$, then plays a crucial role in determining the temperatures of

both regions. This sensitivity is extremely low compared to the global sensitivity of 0.5, resulting in a somewhat strange result of a decreasing polar temperature with increasing area A, even with a slight increase in the net solar radiation in the polar region ($q_2$ in Fig. 3a). Thus, these two parameters ($1/\alpha$ and s) determine the general temperature dependence shown in Fig. 3b. However, there is no rational explanation for this low sensitivity value in this paper or the cited reference (Ou, 2018, Appendix B). Also, I suspect this transfer coefficient applies to air-sea interaction, not air-land interaction. The author invokes an MEP principle for justification, but then the basic logic for the estimation from that principle should be explained.

- *The linkage of differential temperature to differential forcing (I shall call it "local" sensitivity) bears no relation to the "global" sensitivity since latitudinal variation of the LW flux is dominated by that of the convective flux and in fact neglected in our model (see Section 2). This is the reason that the local sensitivity depends only on the air/sea exchange coefficient, which in turn has no import on the global sensitivity. I will add this discussion in the revision.*

- *I will provide an explicit expression of the air/sea exchange coefficient in the revision, which entails Bowen ratio and turbulent wind. Although both vary latitudinally, their effects cancel somewhat (Bowen ratio is smaller in the tropics where wind is also weaker), so we are justified to use their global-means in our minimal model. Setting the global-mean Bowen ratio at 0.25 (Peixoto and Oort 1992, their Fig.14.1) and turbulent wind at 3 m/s (Hartmann 2015, his Fig. 4.7) yields $\alpha = 15\ Wm^{-2}\ ^0C^{-1}$. This estimate is supported (in fact informed) by the observed differential temperature range of $20\ ^0C$ given the equator-to-pole forcing range of $300\ Wm^{-2}$. As a further test, we note that the MEP-deduced MOC (meridional overturning circulation) depends only on $\alpha$, and the above estimate yields a transport of 17 Sv for the North Atlantic, not unlike the observed one. I will add this discussion in the revised paper.*

Minor comments:

1. L104-105: "clouds would self-adjust to stabilize the temperature constrained by intrinsic water properties".

Here, the surface temperature seems to be related to the cloud amount. If so, it would be easier to understand if the cloud amount was also shown in Fig. 1.

- *I will remove this figure since without full closure as discussed in Ou (2001), it raises more questions than it answers, as you have rightfully pointed out. On the other hand, intrinsic water constraint on the surface temperature can be rationalized by simpler statements, such as (to be refined): "surface temperature is bounded below by the greenhouse warming and above by the evaporative cooling due to accelerating rise of the saturation vapor pressure with temperature on account of the Clausius–Clapeyron equation".*

2. Figure 2.

I cannot understand what are the lines shown on the top-right side of this figure. Also, there is a protruding structure in the tropical region. It would be better to explain what these mean.

- *They are the tropopause and polar front dividing the tropical/polar airmasses. The protruding structure is the highland, which is critical in seeding the sea-level ice sheet even when the surface is above the freezing point. I will amend the caption and add this discussion in the text.*

3. L171-172: Equations (3) and (4).

In these equations, the reflectance of the sea surface appears to be assumed to be zero. If so, this assumption should be stated in the text as the sea surface reflectance is known to be about 0.1, which is larger than zero.

- *I shall add a statement about the neglect of the ocean albedo (common practice) in comparison with the land albedo, particularly before the advent of land plants.*

4. L182-183: "the land reflectance is that of a desert set to r = 0.5, ... for a total land area set to 0.3".

Here, the land reflectance (r) seems to be changed for a desert (0.5) and total land (0.3). However, it is unclear how the reflectance is changed in the model calculations shown in Fig. 3. Please resolve this ambiguity.

- *I am sorry about the confusion, which will be clarified in the revision. The land reflectance has a single fixed value 0.5 and the total land (0.3) refers to its area as a fraction of the global surface. The x-axis of Fig. 3 is the tropical land area, which increases from 0 to 0.3.*

5. L217-218: "not ... the ice-albedo feedback".

In this model, the land reflectance is fixed to a value (0.3 or 0.5, see the above comment 4). So this model does not include the ice-albedo effect. If so, one should not deny the ice-albedo feedback by the results obtained from a model that contains no ice-albedo effect.

- *Land reflectance is fixed at one value 0.5, see the preceding response. I will write down the differential heat equation to show that perennial sea ice is nil so long as there is differential SST to institute the OHT (ocean heat transport) maximized by MEP. This is a robust outcome of MEP, which nonetheless may explain why subpolar water remains open during LGM (last glacial maximum) even though it is at the freezing point. Over paleohistory, it is also noted by Eyles (2008) that "no sedimentary evidence of sea ice for all glacio-epochs, contrary to models".*

- *I will expand the discussion of ice-albedo feedback by way of bifurcation diagrams to show its progression from iceball earth to tropical waterbelt to divergent land-ice line and MEP. Since the land-ice line is marked by above-freezing temperature of the co-zonal ocean and sea-ice line has retreated to the pole by MEP, there no runaway ice-albedo feedback until the global ocean is cooled to the freezing point, a precondition of iceball earth.*

- *In addition, I shall argue that the iceball earth is untenable by the global-mean balance since positive greenhouse feedback would warm it to above the freezing point (see my response to Comment 1). It is seen therefore that MEP resolves the faint-young-sun paradox, precludes the iceball earth, and allows tropical glaciation abutting an open ocean.*

6. Figure 3b: the profile of T1.

There is a bend in the temperature profile (T1) in this figure, and I cannot understand why. Could you explain the reason?

- *That is because $T_2$ has already reached the freezing point (the x-axis) and yet its average with $T_1$ equals the global mean (thick solid line). I shall add this statement and elaborate on the differential temperature in discussing the bifurcation diagram.*

7. Equation (9).

I cannot understand the meaning of this equation. Please explain this equation in more detail.

- *This equation is wrong and will be removed.*

*References:*

*Bjordal, J., Storelvmo, T., Alterskjær, K. et al. Equilibrium climate sensitivity above 5 °C plausible due to state-dependent cloud feedback. Nat. Geosci. 13, 718–721 (2020). https://doi.org/10.1038/s41561-020-00649-1*

*Hartmann DL (2015) Global physical climatology. Academic Press, New York, 411 pp*

*Manabe, S. & Wetherald, R. 1967. Thermal equilibrium of the atmosphere with a given distribution of relative humidity. Journ. of Atmosph. Sciences, 24, No. 3: 241-259.*

*Ou HW (2001) Possible bounds on the earth's surface temperature: from the perspective of a conceptual global-mean model. J Clim 14:2976–88 https://doi.org/10.1175/1520-0442(2001)014<2976:pbotes>2.0.co;2*

*Peixoto JP, Oort AH (1992) Physics of Climate. Amer Inst Phys, New York*